# Extracellular phase separation mediates storage and release of thyroglobulin in the thyroid follicular lumen
Yihan Yao [1], Nadia Erkamp [1], Tomas Sneideris [1], Xiqiao Yang[1], Rob Scrutton [1], Matthias M. Schneider [1], Charlotte M. Fischer[1], Erik Schoenmakers[2], Nadia Schoenmakers[2] & Tuomas P. J. Knowles [1] ✉

Thyroid hormones are produced by the thyroid gland and are essential for regulating metabolism, growth and development. Maintenance of circulating thyroid hormone levels within an appropriate range is thus a prerequisite for health. In vivo, this objective is, at least in part, facilitated through an extracellular storage depot of thyroglobulin, the glycoprotein precursor for thyroid hormones, in the thyroid follicular lumen. The molecular basis for how soluble thyroglobulin molecules form such dense depot assemblies remains elusive. Here, we describe in vitro biophysical analysis of thyroglobulin phase behaviour, suggesting that thyroglobulin is prone to undergoing ionic strength-dependent phase separation, leading to the formation of liquid-like condensates. Fluorescence photobleaching measurements further show that these condensates age as a function of time to form reversible gel-like high density storage depots of thyroglobulin. IF experiments on mouse and human thyroid follicles ex vivo reveal that spherical globules of Tg protein dense phase are present in the follicular lumen, consistent with the idea that Tg undergoes phase separation. These findings reveal a molecular mechanism for the last-come-first-served process of thyroglobulin storage and release, suggesting a role for extracellular phase separation in thyroid hormone homeostasis by providing organizational and architectural specificity without requiring membrane-mediated confinement.

Thyroid hormones play an essential role in cell differentiation[1–3], normal body development[4–6], and metabolism[7–10]. Dysregulation of thyroid hormone production and release leads to abnormal blood hormone levels, with adverse sequelae for human health. In particular, thyroid hormone deficiency is associated with impaired growth and neurodevelopment in children[11–13], and has detrimental effects on adult cardiometabolic and neurological health[14–17], while excessive thyroid hormone levels may result in cardiac decompensation[18–20], osteoporosis[21], weight loss[22], and potentially life-threatening thyroid storm[23]. Therefore, stable blood levels of thyroid hormones are crucial for normal physiological function, even when iodine intake is variable. The thyroid gland comprises many structural and functional units called the thyroid follicles. These are spherical structures, enclosed by a virtually impenetrable layer of cuboidal cells, the follicular cells, or thyrocytes, surrounding the follicular lumen[24]. Thyroglobulin (Tg), a large dimeric glycoprotein of 660 kDa, on which thyroid hormone is synthesized, is the most highly expressed protein in the follicular cells of the thyroid gland[25,26]. It is secreted into the follicular lumen through the apical membrane, where it is stored at high concentration (100–600 mg/mL)[27,28] in an extracellular matrix called the colloid which serves as a reservoir for both iodine and Tg[25,29,30].

During hormonogenesis, iodine in the follicular lumen reacts with tyrosine residues on Tg molecules closest to the apical plasma membrane to form mono- and diiodotyrosine (MIT and DIT) which couple to form thyroid hormones triiodothyronine (T3) and thyroxine (T4)[31]. Uptake of iodinated Tg into thyrocytes is mediated by endocytosis[32], followed by proteolytic cleavage of T3 and T4 in the lysosome and secretion into the bloodstream through the basolateral membrane[33]. The iodination and retrieval of Tg from the follicular lumen follows the "last come, first served" rule, i.e., newly synthesised Tg contributes preferentially to hormonogenesis, as it is enriched near the apical membrane and is highly accessible for endocytic internalisation[25,33–35]. Tg molecules are stored in the follicular lumen as covalently linked Tg multimers forming high protein density globules that can be isolated from the surrounding soluble Tg[28,30,36]. High molecular weight Tg multimers only appear after Tg is iodinated, as

[1]Yusuf Hamied Department of Chemistry, University of Cambridge, Cambridge, UK. [2]Wellcome Trust-MRC Institute of Metabolic Science, University of Cambridge Metabolic Research Laboratories, University of Cambridge, Cambridge, UK. ✉e-mail: tpjk2@cam.ac.uk

intermolecular cross-links such as 3-3′ dityrosine bridges occur during Tg iodination[37].

Biomolecular condensates have been investigated extensively in recent decades as a mechanism to dynamically compartmentalise biomolecules in cells[38–41]. Proteins and RNAs can be stored in liquid-like droplets formed through liquid-liquid phase separation at high concentrations, under constant exchange with substances in the cell[38,40,42]. The formation and dissolution of biomolecular condensates are regulated by various cellular factors including protein expression levels, post-translational modifications, chaperones, and enzymes[43–46], providing a dynamic platform for storage and release. Hormones such as insulin have been reported to be stored intracellularly as liquid-like condensates in secretory granules[47,48].

In this study, we demonstrate that human Tg molecules can phase separate in vitro in an ionic-strength-dependent manner, with condensates emerging at high salt concentrations. To obtain insights into the physical properties of Tg condensates, we assessed their sphericity, fusion behaviour, fluidity, and condensate aging[49] in vitro. The reversibility of these gel-like depots is demonstrated in this study, providing molecular insight into the "last come, first served" rule of Tg turnover in the thyroid follicular lumen[25,33–35]. Moreover, we show that Tg-rich globules exhibiting a high degree of sphericity are present in fixed mouse and human thyroid follicular lumen. The combined findings from in vitro and ex vivo experiments suggest that extracellular phase separation of Tg facilitates its storage and release in the thyroid follicular lumen. As Tg serves as the platform for thyroid hormone production, its storage and release processes can have a significant impact on regulating thyroid hormone homoeostasis.

## Results

### Thyroglobulin undergoes phase separation in vitro in an ionic strength-dependent manner

With increasing evidence suggesting a role for phase separation in facilitating the biogenesis of secretory storage granules of insulin[47], we hypothesised that Tg may also undergo phase transition as part of its biological function during storage and release. We first set out to probe whether Tg can phase separate in vitro. For this purpose, Tg was labelled with Alexa Fluor 488 via amide coupling. Given the high concentration of iodide in the thyroid follicular lumen, we explored the phase separation potential of human Tg by varying the concentration of both Tg and KI or KCl, with the addition of polyethylene glycol (PEG) 20k, a commonly used biocompatible crowding reagent mimicking the densely packed thyroid follicular lumen[50]. Fluorescence confocal microscope imaging revealed Tg condensate formation above a critical salt and protein concentration threshold (Fig. 1A, B) and with higher salt concentration, the critical protein concentration required for Tg phase separation becomes lower. To gain deeper insight into the effects of protein and salt concentrations on Tg phase separation behaviour, we used a high-throughput combinatorial microfluidic platform to generate a high-resolution phase diagram of [Tg] against [KCl][51]. Approximately 12,000 data points representing 12,000 unique microenvironments were collected using this platform, and the resulting phase boundary delineates the ionic strength-dependent character of Tg phase separation (Fig. 1C). According to the phase diagram, Tg phase separation occurs above 100 mM KCl, and condensates can be observed with less than 2 μM Tg in the phase-separating system. Phase-separation experiments with unlabelled Tg showed a similar behaviour (Supplementary Fig. 1), suggesting that the Alexa Fluor 488 label does not affect the phase-separation behaviour of Tg.

### Thyroglobulin phase separation results in liquid-like condensates that age over time

In vitro experiments confirm that Tg phase separates under physiologically relevant conditions. However, the physical properties of Tg condensates remain unknown. One prominent characteristic of liquid-like condensates is that they adopt a spherical shape to minimise the interface energy. Indeed, spherical condensates form at different Tg protein concentrations (Fig. 2A). Notably, the dense phase fraction increases with higher initial Tg

concentration, conforming to a biphasic phase diagram[38]. We then quantified the roundness (the inverse of the aspect ratio; ranges between 0 and 1, with 1 representing the perfect sphere) of condensates formed at 5 μM, 10 μM, and 15 μM Tg. According to the distribution plot of condensate roundness, the majority of Tg condensates adopt a roundness value between 0.8 and 1.0 at all three protein concentrations, indicating a high degree of sphericity (Fig. 2B).

Further validation of the liquid-like nature of Tg condensates comes from the observation of their ability to fuse, implicating high mobility of Tg molecules in the condensates (Fig. 2C), another important characteristic of liquid-like condensates[52]. The high mobility of Tg molecules within the condensate was further confirmed using fluorescence recovery after photobleaching (FRAP) assays carried out on freshly formed Tg condensates in vitro, showing recovery of more than half of the original fluorescence signals within ~80 s (Fig. 2D). Condensates thus have a liquid-like nature directly after their formation. In contrast, FRAP on Tg condensates incubated for 1 h at room temperature showed little fluorescence recovery (Fig. 2E). This indicates that the material properties of Tg condensates change over time, with a decrease in the molecular dynamics, possibly taking a more gel-like condensate content, mediating its transition from soluble molecules into more dense deposits.

### Thyroglobulin condensates convert to reversible gels

For Tg condensates to function as a high-density storage depot, molecules need to be released efficiently. To investigate this, Tg condensates were formed in a solution of 250 mM KI, 4% w/v PEG 20k, and 50 mM HEPES buffer at pH 7.5. Within 1 min after condensate formation, we diluted the sample to 50% of the original concentrations, leading to the dissolution of all the condensates within 10 min (Fig. 3A). Essentially, we moved from the phase separating region to the mixed region of the phase diagram. In response, condensates dissolved, and the system became well-mixed. This shows that Tg molecules can be released from freshly formed condensates. Notably, the dissolution initiates at the centre of Tg condensates, and gradually extends outward. This observation suggests that the aging of condensates is initiated at the interface to the surroundings, as has been previously reported for FUS condensates[53,54]. FRAP experiments on partially aged Tg condensates further revealed faster fluorescence recovery at the condensate centre compared to the interface (Supplementary Fig. 2), providing additional evidence that aging initiates at the condensate interface. Next, we performed a similar dilution experiment on Tg condensates 1 h after formation. These Tg condensates wetted on the bottom glass surface of the imaging chamber after the 1-h incubation at room temperature (Fig. 3B, before dilution), suggesting aging of condensates. Most of the Tg dense area dissolved within 10 min after buffer addition (Fig. 3B). Thus, while the material properties of condensates may change over time, they could still be dissolved. Overall, the reversibility of Tg phase separation establishes a dynamic reservoir for rapid protein storage and release.

### The spatial distribution of Tg in cultured mouse and human thyroid follicles is characteristic of phase separation in vivo

Our in vitro experiments strongly indicate that Tg forms ionic strength-dependent liquid-like condensates, which can be dissolved to release the stored Tg molecules. Next, we performed ex vivo immunofluorescence (IF) experiments on cultured mouse and human thyroid follicles to further explore if Tg phase separates in vivo. In order to distinguish the lumenal space from the rest of the follicle, thyrocyte nucleic DNA was stained with DAPI; the apical aspect of thyrocytes was immunostained with anti-ezrin, a cytoskeleton protein, delineating the apical membrane enclosing the thyroid lumen[55] and Tg was immunostained with AF594 conjugated antibodies (Fig. 4A, B). In three independent sets of cultured primary mouse thyroid follicles and two independent sets of cultured primary human thyroid follicles, protein-dense Tg globules could be observed in the follicular lumen. Quantification of the roundness of the globular Tg dense phases demonstrates that they adopt a high degree of sphericity, with roundness values exceeding 0.75 for all analysed globules (Fig. 4 roundness). This observation

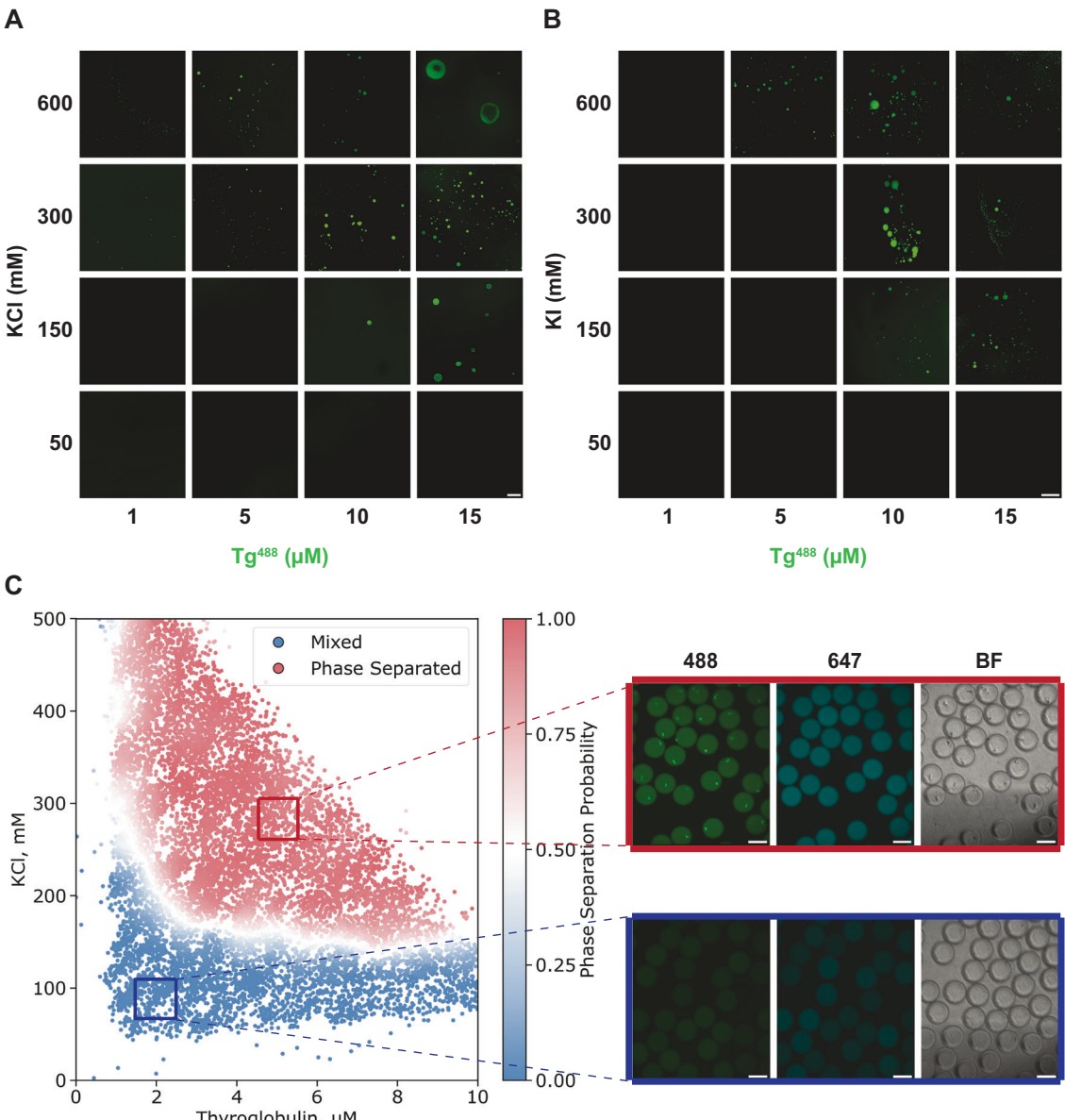

**Fig. 1 | The phase separation of Tg in vitro is ionic strength dependent. A** A phase diagram of [Tg] against [KCl] illustrated by confocal images taken under a range of Tg and KCl concentrations. Scale bar, 50 μm. **B** A phase diagram of [Tg] against [KI] illustrated by confocal images taken under a range of Tg and KI concentrations. Scale bar, 200 μm. Images are representative images. $n = 3$ per experimental condition for (**A**) and (**B**). **C** A [Tg] against [KCl] phase diagram acquired using PhaseScan (left), with images of droplets in microfluidic channels (right) within the phase separation regime (red box) and the mixed regime (blue box), under 488 nm wavelength (Tg protein), 647 nm wavelength (KCl) fluorescence channels and bright field. The white curve between the two regimes indicates the phase boundary. $n = 11773$. Scale bars, 100 μm. All samples contained 4 w/v% PEG 20k in 50 mM HEPES buffer pH = 7.5.

suggests a key role for surface tension in defining the shape of the storage depots early in their formation, consistent with the idea of phase separation into a dense liquid. There is little fluorescence signal of ezrin or Tg in the isotype control experiments, confirming the high specificity of ezrin and Tg immunostaining (Supplementary Fig. 3). These observations from ex vivo experiments confirm the presence of Tg protein-dense phases with a high degree of sphericity in the thyroid follicular lumen from both mice and humans, characteristics which emerge from surface tension-mediated forces, and are thus indicative of the involvement of a dense liquid state in the formation of the storage depots.

## Discussion

In this study, we report that Tg undergoes ionic strength-dependent phase separation and forms reversible condensates in vitro. Using PhaseScan, a high-resolution phase diagram of Tg against KCl concentration reveals the impact of salt on Tg phase separation. Phase separation of Tg is promoted at high KCl or KI concentrations, with lower protein concentration required for Tg phase separation under higher salt conditions. The high degree of sphericity, coalescence, and rapid recovery of fluorescence signals after photobleaching of a small area in freshly-formed Tg condensates provide concrete evidence for their liquid-like nature. Tg condensates age over time, leading to reversible gel-like storage depots. As the thyroid follicular lumen contains iodide at a high concentration[56], it is highly likely that Tg phase separates in this high salt extracellular environment. IF ex vivo experiments on Tg molecules in mouse and human thyroid follicle samples further revealed spherical globules of protein-dense area, further indicating the phase separation potential of Tg in vivo.

Our findings demonstrate that extracellular phase separation of Tg is a possible mechanism to mediate stable, high-concentration protein storage and release in the thyroid follicular lumen. Our experimental results

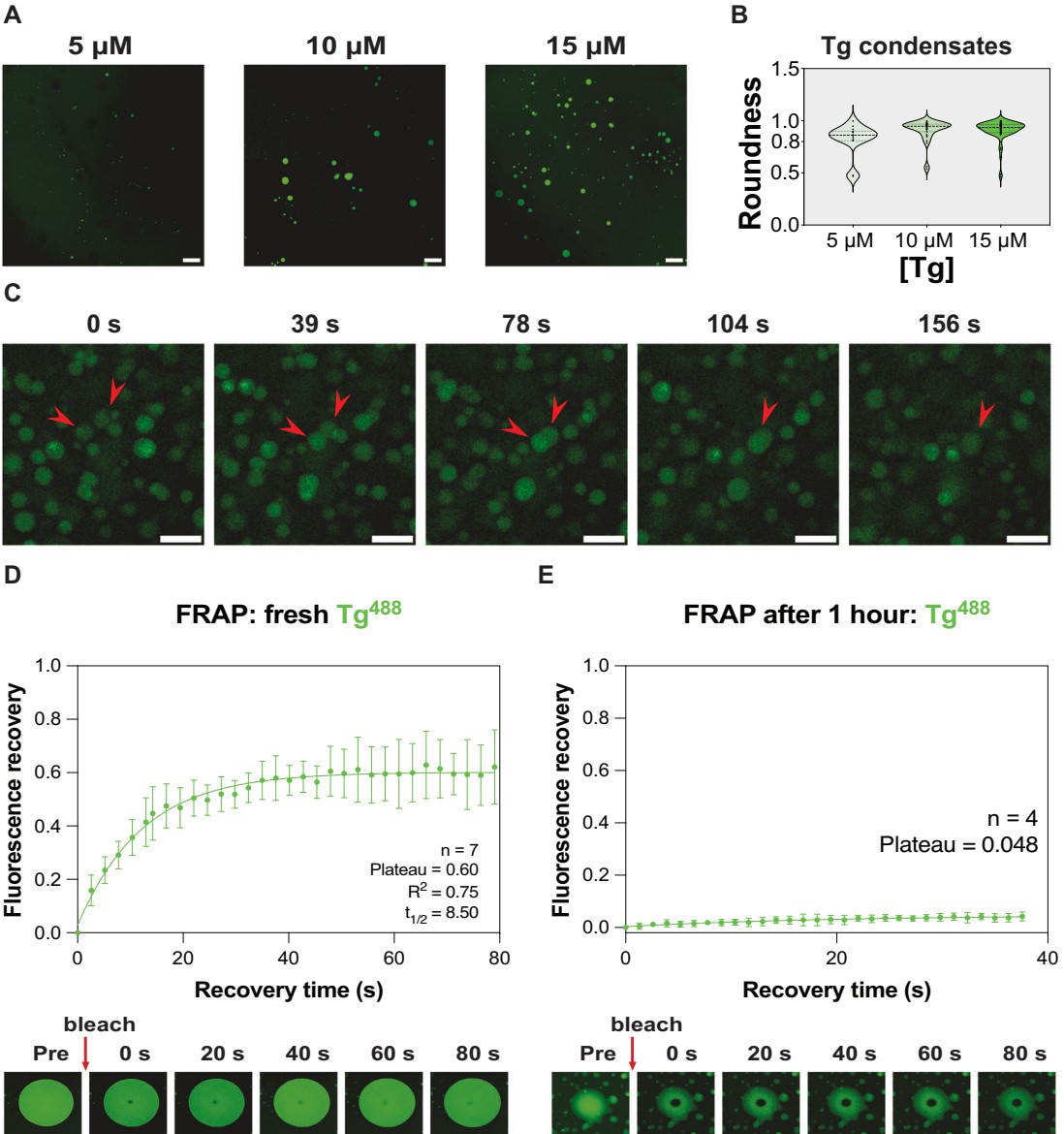

**Fig. 2 | Tg can form liquid-like condensates in vitro that age over time. A** Confocal images of Tg$^{Alexa Fluor 488}$ condensates formed in vitro at 5 µM, 10 µM, and 15 µM Tg. Scale bars, 100 µm. **B** Roundness analysis of Tg condensates formed at 5 µM, 10 µM, and 15 µM Tg. The median, 25th percentile, and 75th percentile are shown as dotted lines in the violin plot. Roundness = $4 \times$ area/($\pi \times$ major_axis$^2$). Number of data points $n = 14, 31, 60$ at 5 µM, 10 µM, and 15 µM Tg respectively. Samples contained 300 mM KCl and 4% w/v PEG 20k. **C** Fusion behaviour of Tg condensates over time (indicated with red arrow heads). Condensates were formed under 10 µM Tg, 250 mM KCl, and 4% w/v PEG 20k. Scale bars, 10 µm. **D** FRAP analysis of freshly formed Tg condensates and **E** 1-h aged Tg condensate in vitro. Error bars represent standard deviation. Scale bars, 10 µm. Both fresh and aged condensates were formed at 13.75 µM Tg, 250 mM KCl, and 4 w/v% PEG 20k. For all experiments in vitro, 50 mM, pH = 7.5 HEPES buffer was used.

combined with previous reports[28,30,31,35,36] suggest that Tg (orange dots) initially phase separates to form liquid-like condensates after it is synthesised and secreted by thyrocytes into the thyroid follicular lumen. Tg condensates age over time, leading to the formation of a reversible gel. Finally, multi-merised Tg storage depots form with an outer layer composed of newly-synthesised Tg (red dots) ready to be chemically modified and released[25]. Thyrocytes then retrieve chemically modified Tg via endocytosis, and thyroid hormones T3 and T4 (blue dots) are liberated from Tg polypeptide backbone during proteolysis before being secreted into the bloodstream[25] (Fig. 5).

In this study, we have not probed the regulation of Tg storage and release, which is known to be controlled by hormonal (e.g., TSH) and enzymatic (e.g., cathepsin-mediated) means. Interestingly, however, studies suggest that TSH stimulates the turnover of newly synthesised Tg at a higher rate than older Tg stores[57], consistent with the last-come-first-served model that emerges from the condensate picture. Furthermore, regulation of other types of biomolecular condensates through enzymes and post-translational modifications has been described, and this could offer a route to further influence the rate of release of Tg[43–46].

Overall, the findings suggest that the extracellular phase separation of Tg is a possible mechanism providing structural and organisational specificity without the need for membrane boundaries to mediate high-concentration protein storage and release in the thyroid follicular lumen and is coherent with the current knowledge of thyroid functioning. While protein phase separation has garnered extensive study inside the cell, limited research has delved into its role in the extracellular environment. Previous studies have demonstrated the role of extracellular phase separation in the assembly of extracellular matrix proteins[58]. Our report introduces a novel

**Fig. 3 | Tg condensates formed in vitro can be dissolved by dilution. A** The dissolution of freshly formed Tg condensates over time after reducing concentrations by half. Buffer was added into the system at 0 s, within 1 min after Tg condensate formation. **B** The dissolution of 1-h aged Tg condensates over time after reducing concentrations by half. Buffer was added into the system at 0 s, 1 hr after condensate formation. In both A and B, Tg condensates formed at 10 μM Tg, 250 mM KI, and 4% w/v PEG 20k. 50 mM, pH = 7.5 HEPES buffer was used and the total volume of the phase separating system was 10 μL. At 0 s, 10 μL buffer was added into the system. Representative images, *n* = 3 independent experiments. Scale bars, 100 μm.

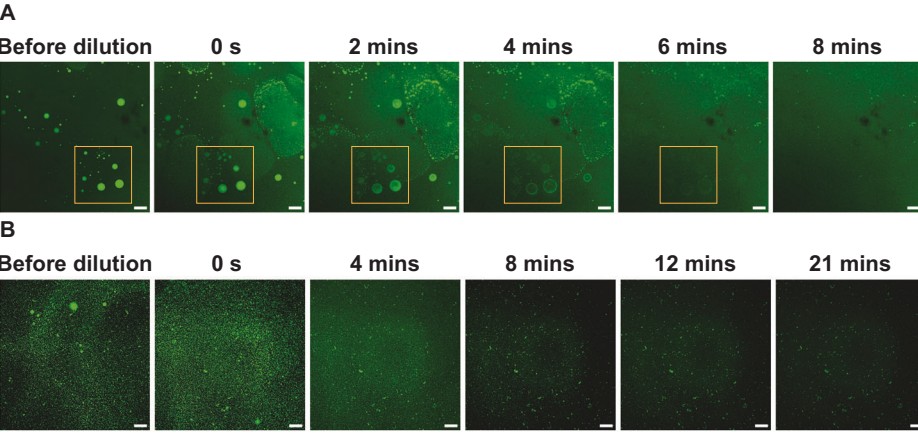

## Methods

### Materials

All reagents and chemicals were purchased with the highest purity available. Human thyroglobulin protein (lyophilised powder), PEG 20k, 1,6-hexanediol, and the buffer reagent HEPES were from Sigma-Aldrich. KCl was from Fisher Scientific. AlexaFluor488 NHS ester (A20000) and AlexaFluor647 carboxylic acid, tris(triethylammonium) salts (A33084) were purchased from ThermoFisher Scientific. 008-FluoroSurfactant was purchased from Biotechnologies. HFE-7500 3M was from Fluorochem. For the Tg IF experiment: primary anti-Tg antibody: Abcam rabbit anti-TG Ab156008; secondary antibodies: Abcam Donkey pAb to Rabbit IgG Fluor 594 Ab150064; for the isotype control: Abcam anti-rabbit IgG 172730.

### Microfluidic device design and fabrication

The device design and fabrication protocol were performed as previously described in Arter, W. E. et al[51].

Poly(dimethylsiloxane) (PDMS)-based microfluidic devices are made for the PhaseScan experiment. Microfluidic devices were fabricated by photo lithography and soft lithography. Firstly, AutoCAD (AutoDesk) was used to design microfluidic devices and then these designs were printed on a photomask (Micro Lithography). SU8-3050 photoresist (Microchem) was poured onto a polished silicon wafer and spinned at 1500 rpm for 15 s and then 3000 rpm for 45 s. Then the pattern of the mask was transferred by UV exposure (60 s) to the silicon wafer coated with a layer of SU8-3050 photoresist. Excessive SU-8 photoresist was removed using propylene glycol methyl ether acetate (PGMEA; Sigma). The wafer with SU-8 patterns (i.e., master) was dried by blowing with nitrogen and baking at 95 °C on a hot plate. The master was placed in a plastic petri dish and served as a mould for (PDMS; Sylgard184, Dow Corning) casting. The PDMS base and cross-linking reagent were mixed in a 10:1 mass ratio and polymerised by baking at 65 °C for 2 h in the oven. The PDMS device was cut from the petri dish and cleaned by isopropanol. Isopropanol on the PDMS replica was dried by blowing with nitrogen and putting in 65 °C oven for 3 min. Both the PDMS replica and the glass slide were activated in an oxygen plasma oven (30 s, 40% power, Femto, Diener Electronics) before bonding. The channels were treated with 1% (v/v) trichloro(1H,1H,2H,2H-perfluorooctyl) silane (Sigma, 7.5 μL) in HFE-7500 (Fluorochem, 495 μL) for 1 min, before being dried with nitrogen gas and heated at 95 °C for 3 min.

### PhaseScan

The experimental procedure for preparing phase diagrams using PhaseScan is similar to the previous description in Arter, W.E. et al[51]. Instead of using syringe pumps to control the flow rate of input solutions, pneumatic pumps (Fluigent) were employed for more precise control of flow rates. The microfluidic device used for Tg PhaseScan experiment is shown in Supplementary Fig. 4A. Four aqueous solutions were loaded into four 2 mL Eppendorf tubes connected to four pneumatic pumps. The aqueous stock solutions and their flow conditions were: KCl stock solution: 1 M KCl in 50 mM HEPES pH = 7.4, with 6 μM AF647 carboxylic acid; variable flow rate at 5–40 μL/h. Final [KCl] in droplets ranges from 83 to 666 mM. Thyroglobulin stock solution: 25 μM thyroglobulin in 50 mM HEPES buffer pH = 7.4, 50% of Tg was 1:1 (molar ratio of dye:protein = 1:1 when labelling the protein sample overnight at 4 °C) labelled with AF488 NHS ester; variable flow rate at 5–40 μL/h. Final [Tg] in droplets ranges from 2.08 to 16.67 μM. Buffer: 50 mM HEPES pH = 7.4; variable flow rate at 5–40 μL/h. PEG: 24% w/v PEG 20 K in 50 mM HEPES pH = 7.4; constant flow rate at 10 μL/h; concentration of PEG 20k in droplets: 4% w/v. The total aqueous flow rate is constant at 60 μL/h. The oil used for the experiment was HFE-7500 3 M with 2% 008-FluoroSurfactant. Droplets were imaged when flowing in the imaging chamber of the microfluidic device (Supplementary Fig. 4A) under a constant oil flow rate of around 120 μL/h.

### Sample preparation and imaging of Tg condensates formed in vitro

Tg was labelled by AlexaFlour488 NHS ester. The labelling was performed at a molar ratio of 1:1 (dye:protein) overnight at 4 °C. In vitro experiments on Tg condensate were prepared by manually pipetting KCl/KI, PEG 20k, and Tg labelled with AlexaFlour488 NHS ester into the Eppendorf™ Protein LoBind tube. The total volume of the mixture was kept constant at 10 μl. The buffer used was pH = 7.5, 50 mM HEPES buffer; KCl/KI, PEG 20k, and the protein were dissolved in the pH = 7.5, 50 mM HEPES buffer. For imaging, the mixture was transferred from the Protein LoBind tube onto an imaging chamber (Supplementary Fig. 4B). The imaging chamber was made by covalently attaching PDMS onto a glass coverslip after activation in the oxygen plasma oven. During imaging, the chamber was sealed by a round glass coverslip to prevent water evaporation and concentration change. A Leica Stellaris 5 confocal microscope equipped with a 10× Nikon 0.3 NA or a 63× oil immersion Leica 1.4 NA objective was used for imaging. Laser of 488 nm wavelength and bright field were used for imaging.

### FRAP assays on in vitro Tg condensates

FRAP assays were carried out on condensates composed of AF488 NHS ester labelled Tg. Tg condensates for FRAP were formed under 13.75 μM Tg, 250 mM KCl, and 4% w/v PEG 20k. 50 mM, pH = 7.5 HEPES buffer was used and the total volume of the phase separating system was 10 μl, placed in the imaging well made by PDMS (Supplementary Fig. 4B). The aged sample was prepared by placing the same phase separating system at room temperature for 1 h before FRAP experiments. The chamber containing samples was sealed by a round glass coverslip to prevent water evaporation and concentration change. A Leica Stellaris 5 confocal microscope equipped

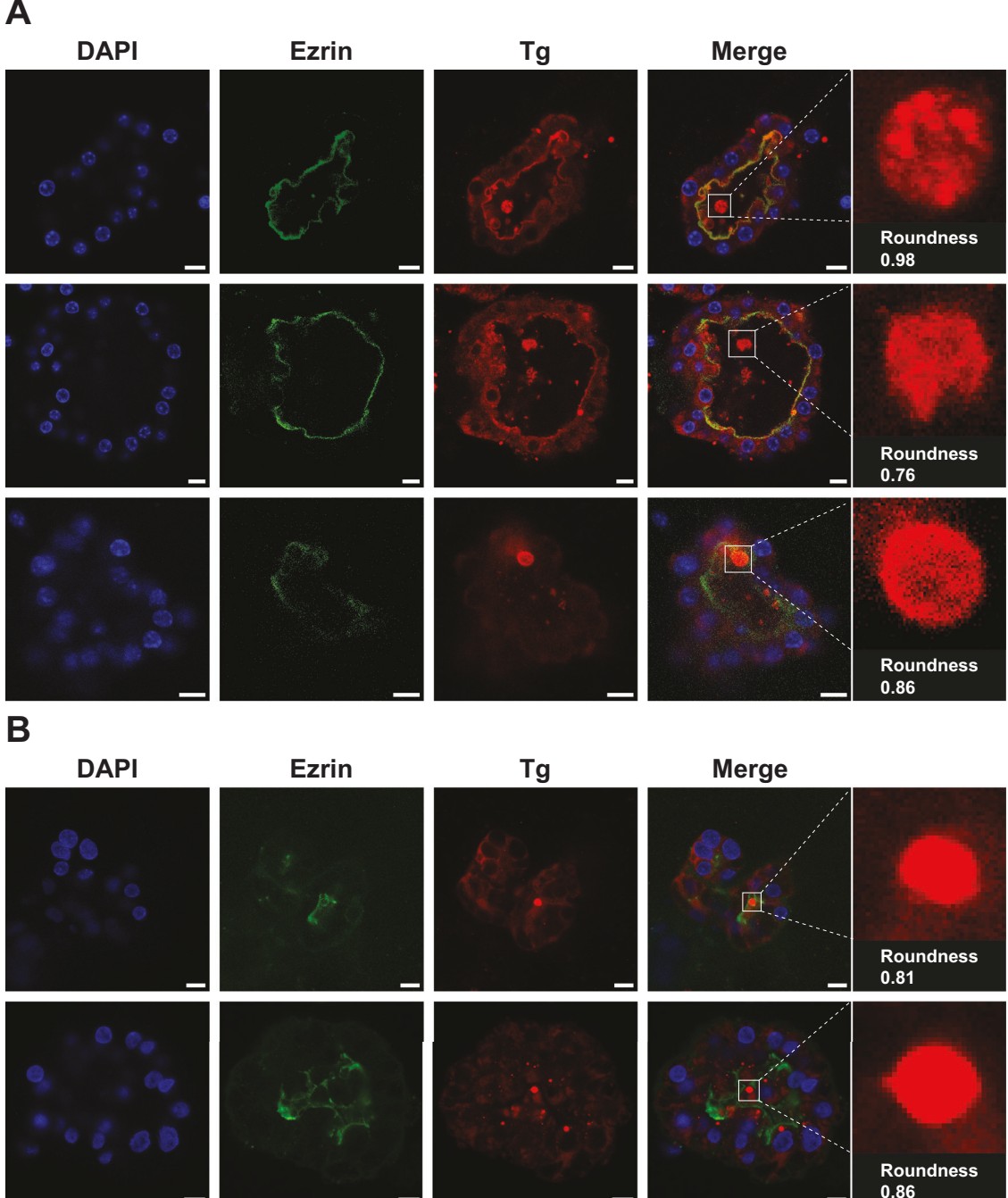

**Fig. 4 | Tg shows a heterogeneous distribution in cultured mouse and human thyroid follicular lumen, with observation of globules of protein dense phase. A** Confocal images of three thyroid follicles from three independent sets of cultured primary mouse thyroid samples, representing three biological replicates. **B** Confocal images of two thyroid follicles from two independent sets of cultured primary human thyroid samples (from patients with treated Graves' disease), representing two biological replicates. In both (**A**) and (**B**), follicles were imaged under DAPI (blue; thyrocyte nuclei), AF488 (green; ezrin), and AF594 (red; Tg) fluorescence channels. Red globular Tg dense phases inside the lumen are zoomed in for quantifying roundness. Scale bars, 10 μm.

with a 10× Nikon 0.3 NA objective was used for FRAP assays, using the 488 nm laser line. In FRAP experiments described in Fig. 2D, E, 50% laser power was used for bleaching. In FRAP experiments described in Supplementary Fig. 2, 100% laser power was used for bleaching. Image analysis was performed with ImageJ2 (version 2.14.0/1.54f).

### Immunofluorescence experiments on cultured mouse thyroid follicles

We have complied with all relevant ethical regulations for animal use. Thyroid tissue was obtained from surplus wild-type mice (C57BL/6) bred under different protocols under the Animals (Scientific Procedures) Act 1986 Amendment Regulations 2012 following ethical review by the University of Cambridge Animal Welfare and Ethical Review Body and shared amongst facility users according to 3Rs. Animals were housed in individually ventilated cages in an environmentally controlled room, temperature 22 °C (±2 °C) and 55%RH (±10%), with a 12/12-h light/dark cycle, and ad libitum access to food and water. The study used pooled samples from C57BL/6 mice, $n = 10$ females, $n = 1$ male, aged 5–47 weeks. Mice were culled humanely using a rising concentration of $CO_2$ and the thyroid was dissected and digested with 1 mg/mL collagenase and dispase (Sigma

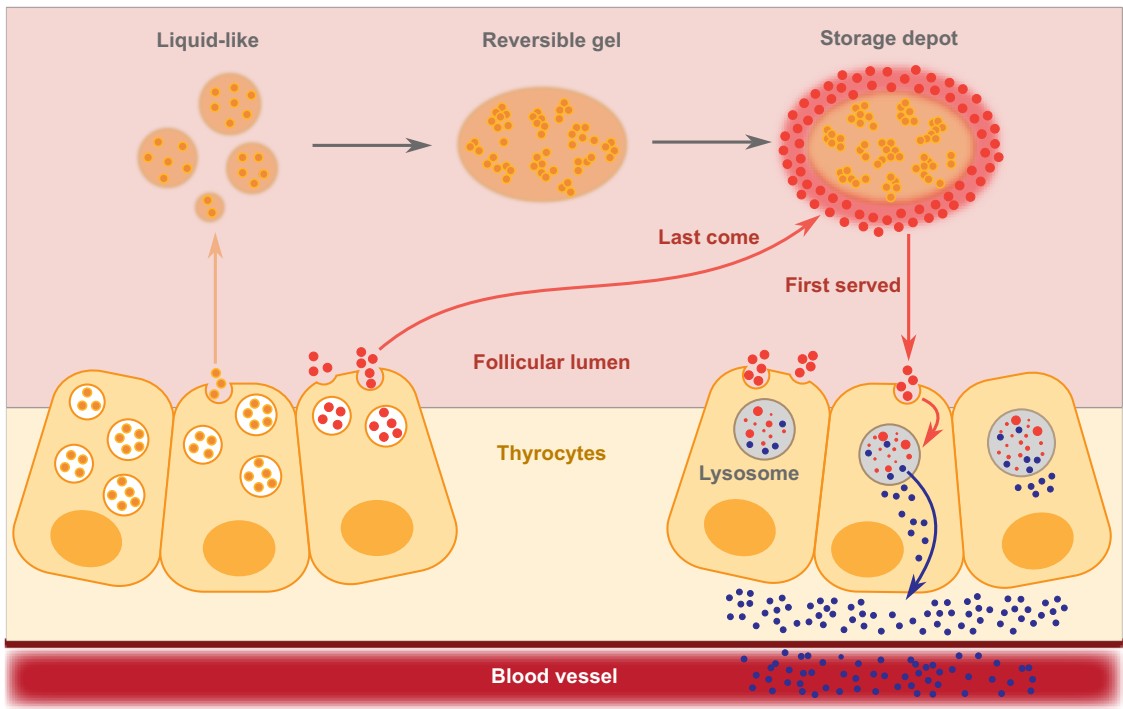

**Fig. 5 | Summary scheme of phase separation in thyroglobulin (Tg) storage and release in the thyroid follicular lumen.** Tg (orange dots) is synthesised and secreted by thyrocytes into the thyroid follicular lumen and they phase separate into liquid-like condensates, which age and become a reversible gel over time, storing Tg at high concentration in depots. Newly-synthesised Tg molecules (red dots) reside on the outer layer of storage depots, near the thyrocyte apical membrane, susceptible to chemical modification and subsequent retrieval by thyrocytes through endocytosis. Within thyrocytes, thyroid hormones (blue dots, T3, and T4) are released from the Tg polypeptide backbone through proteolysis before being secreted into the bloodstream.

C2764, Gibco 17105041). Thyroid follicles were resuspended in Cultrex Reduced Growth Factor Basement Membrane Extract, Type 2 (3533-010-02 R&D systems) and seeded on glass chamber slides in 20 μL domes and cultured in F-12 Ham Nutrient media with 1% penicillin/streptomycin, 10% FBS, 5 μg/mL Human apotransferrin (Sigma T2252), 3.5 ng/mL Hydro-cortisone (Sigma 0396), 2 ng/mL Glycyl-L-histidyl-L-lysine acetate (Sigma 1887), 10 ng/mL Somatostatin (Sigma 1763), 10 μg/ml human insulin (Sigma 19278) and 500 ng/mL bovine TSH (Sigma T8931). Historically, the activity of this product was determined to have a range of 1.5–2.5 IU/mg protein (Sigma). Follicles were cultured for a maximum of 5 days then fixed (4% PFA), permeabilized (0.5% Triton X100), and blocked (5% Bovine serum albumin). Immunostaining was performed overnight using Primary antibodies (anti-thyroglobulin Abcam Ab156008, 1:50 and anti-Ezrin, Invitrogen clone 3C12 1:100 or equivalent concentrations of isotype control; Abcam anti-rabbit IgG 172730 and Invitrogen anti-mouse IgG1 kappa 14-4714-81). After washing, samples were incubated overnight with secondary antibodies (Abcam Donkey pAb to Rabbit IgG Fluor 594, Ab150064 1:500, and Invitrogen donkey anti-mouse Alexa Fluor 488 A21202 1:150) and protected from light. Following nuclear staining with 1:1000 DAPI, samples were imaged with a Leica SP8 lightning confocal microscope equipped with a 63× oil immersion Leica 1.4 NA objective using lasers of 350 nm (DAPI), 488 nm (AF488), and 594 nm (AF594) wavelength and bright field.

### Immunofluorescence experiments on cultured human thyroid follicles

All ethical regulations relevant to human research participants were followed. Human participants provided written informed consent to participate in this study. Two thyroidectomy samples were collected under the auspices of an ethically approved study (21/SC/0158) by South Central Oxford B REC from 2 female patients having routine thyroidectomy for Graves' disease. Both individuals were on antithyroid medication (case 1; carbimazole, Lugol's iodine and iopanoic acid, case 2; carbimazole and

levothyroxine). Tissue was digested in 0.2% Type 2 collagenase (Gibco 17101015) and then washed several times in HBSS prior to resuspension in Cultrex Reduced Growth Factor Basement Membrane Extract, Type 2 (3533-010-02 R&D systems) and seeding on glass chamber slides in 20 μL domes. Cells were initially cultured in Coon's F-12 modified liquid medium (with 2.5 g/L NaHCO3, without L-glutamine) [Biochrom—F0855] with 1% L-Glutamine [Invitrogen—25030024], 1% Penicillin/streptomycin, 4% FBS and 150 ng/mL bovine TSH (Sigma T8931) and 0.4 μg/mL human insulin (Sigma 19278). After 3 days, the medium was changed to serum-free and TSH-free but still containing 1% L-Glutamine, 1% Penicillin/streptomycin, and 0.4 μg/mL human insulin. TSH was added at a concentration of 5000 ng/mL after a further 4 days and cells were fixed for immunostaining 4 days later. Immunostaining and confocal imaging follow the same protocol as mouse samples.

### Statistics and reproducibility

All microscopy images shown are representative of at least three independent experiments. Tg condensate merging experiment was repeated three times with similar results as shown in Fig. 2C. The curve fittings for the FRAP experiment, shown in Fig. 2D, E, were derived from seven and four independent FRAP experiments, respectively. The curve fitting for the FRAP experiment shown in Supplementary Fig. 2 was derived from five independent FRAP experiments for both the centre and interface of Tg condensates. Tg condensate dissolution experiments shown in Fig. 3 were repeated 3 times with similar results for both fresh and aged Tg condensates. FRAP analysis (Fig. 2D, E and Supplementary Fig. 2) and the distribution plot of the roundness of Tg condensates (Fig. 2B) were performed and plotted with GraphPad Prism (Version 10.4.1). IF experiments were carried out on three independent sets of cultured primary mouse thyroid samples with similar results shown in Fig. 4A; IF experiments were carried out on two independent sets of cultured primary human thyroid samples with similar results shown in Fig. 4B. The confocal images of each thyroid follicle

in Fig. 4 are representative of thyroid follicles within each independent set of cultured primary mouse and human thyroid samples.

## Image analysis and plotting

Acquired images from PhaseScan were analysed by a Python (Python version 3.9.7) script[51]. Both the analysis of Tg condensates formed in vitro and the roundness analysis of Tg dense phases in mouse and human thyroid follicles were performed by ImageJ2 (version 2.14.0/1.54 f). Roundness = $4 \times area/(\pi \times major\_axis^2)$. Adobe Illustrator® is used to generate the final figures shown.

## Reporting summary

Further information on research design is available in the Nature Portfolio Reporting Summary linked to this article.

## Data availability

All data generated or analysed during this study are included in this article and its supplementary information files. Original images and data will be made available upon request. Supplementary Data 1 (Source Data.xls) are provided with this paper.

## Code availability

The custom codes used in the current study are available at GitHub: https://github.com/rqi14/PhaseScan.

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

## Acknowledgements

The research leading to these results has received funding from Cambridge Trust and China Scholarship Council (Y.Y.), a Royall Scholarship (N.A.E.), the European Union's 279 Horizon 2020 research and innovation programme under the Marie Skłodowska-Curie grant MicroREvolution 280 (agreement no. 101023060; T.S.), the Newman Foundation (T.S., M.M.S., T.P.J.K.), the European Research Council 437 under the European Union's Seventh Horizon 2020 research and innovation program through 438 the ERC grant DiProPhys (agreement ID 101001615, C.M.F.), Transition Bio (R.S.), the Wellcome Trust Collaborative 283 Award 203249/Z/16/Z (T.P.J.K.) and the European Research Council under the European Union's 285 Seventh Framework Programme through the ERC grants PhysProt (T.P.J.K., agreement no. 337969; 286). N.S. and E.S. are supported by a Wellcome Senior Fellowship (219496/Z/19/Z) or Investigator (210755/Z/18/Z) awards respectively and NIHR Cambridge Biomedical Research Centre (NIHR203312). The views expressed are those of the authors and not necessarily those of the NIHR or the Department of Health and Social Care. The Cell and Tissue Imaging Core in the IMS-MRL is supported by the Medical Research Council (grant number MC_UU_00039). We acknowledge staff in the Anne McLaren Building, Cambridge Biomedical Campus, for animal husbandry.

## Author contributions

Y.Y., N.A.E., C.M.F., and T.P.J.K. conceived the study. Y.Y., N.A.E., T.S., X.Y., M.M.S., N.S., and E.S. performed experiments. R.S., Y.Y,. and T.S. analysed and interpreted the data. Y.Y. and T.P.J.K. wrote the original draft of the paper. T.P.J.K. acquired funding. All authors reviewed and edited the paper.

## Competing interests

The authors declare no competing interests.
