## [Transparent Peer Review file · Communications Biology]

Extracellular phase separation mediates storage and release of thyroglobulin in the thyroid follicular lumen

Corresponding Author: Professor Tuomas Knowles

This manuscript has been previously reviewed at another journal. This document only contains information relating to versions considered at Communications Biology.

Version 0:

Reviewer comments:

Reviewer #1

(Remarks to the Author)

Thyroid hormones are produced by the thyroid gland and are essential for regulating metabolism, growth, and development. Maintaining circulating thyroid hormone levels within a healthy range is critical for overall health. The thyroid gland achieves this, in part, by utilizing an extracellular storage depot of thyroglobulin (Tg), a glycoprotein precursor for thyroid hormones. However, the molecular mechanisms underlying the assembly of soluble thyroglobulin molecules into dense storage depots remain unclear.

This study provides an *in vitro* biophysical analysis of thyroglobulin phase behavior, revealing that Tg undergoes ionic strength-dependent phase separation, forming liquid-like condensates at high salt and protein concentrations. The phase separation behavior is modulated by salt concentration, with higher ionic strength reducing the protein concentration required for condensation. Notably, Tg condensates are liquid-like upon formation, as evidenced by their high degree of sphericity, fusion behavior, and fluorescence recovery after photobleaching (FRAP). However, these condensates age over time, transitioning to a gel-like state with diminished molecular mobility. Remarkably, aged Tg condensates can revert to a soluble state upon dilution.

The manuscript also demonstrates the presence of protein-dense Tg globules in the follicular lumen of cultured primary mouse and human thyroid follicles, which display a high degree of sphericity. This observation supports the hypothesis that Tg undergoes phase separation *in vivo*, playing a critical role in thyroid hormone homeostasis.

The authors suggest that the extracellular phase separation of Tg provides a dynamic and reversible storage mechanism, linking molecular biophysics to endocrine physiology. The findings are novel and offer insights into how Tg storage and release processes are regulated without requiring membrane-bound compartments. However, the following comments should be addressed to enhance the manuscript:

1. In the condensate dissolution experiment (Fig. 3), the dissolution starts at the center. Could the authors conduct FRAP experiments with the laser focused on different regions within the condensate (as shown in Fig. 2D) to determine molecular dynamics at the center versus near the surface? This could help explain why dissolution begins at the core (and aging starts at the periphery).
2. What is the size distribution of aged condensates *in vitro*? Would it be possible to measure FRAP at different positions within larger aged condensates? The example in Fig. 2E appears too small for such analysis, but larger condensates may allow more detailed spatial FRAP studies (as pointed in 1 above).
3. Is there evidence of fibril-like structures forming within aged condensates (at least *in vitro*)? The depiction in Fig. 5 suggests fibrils in the interior, but direct evidence is lacking.
4. Fig. 4A shows a significant accumulation of Tg on the thyrocyte apical membrane. Could this represent newly synthesized Tg molecules? Would repeating this experiment over time reveal dynamic changes in condensate size or number, or are Tg concentrations expected to remain relatively stable at the membrane and within the condensates?
5. Is there any other evidence of the physico-chemical properties of the condensates in Fig 4 besides roundness?
6. In Fig. 5, the depiction of a Tg outer layer around the condensate during the transition from a reversible gel to a storage depot is not directly supported by the data. While the *in vitro* experiments (Figs. 1–3) and *ex vivo* observations (Fig. 4) establish condensate formation and presence in the lumen, there is no direct evidence of a distinct Tg outer layer. Can the authors clarify or provide justification for this proposal?
7. Similarly, the mechanistic step in Fig. 5 depicting the relocation of newly synthesized Tg from the apical membrane to the

condensate (via the "last come" red arrow) lacks experimental support. Wouldn't it make more sense for Tg to be recruited directly from the condensate itself, as demonstrated in the dissolution experiment (Fig. 3)? What is the functional role of this proposed Tg outer layer? Could the condensate alone serve as the storage depot?

8. Do the authors believe this mechanism of extracellular phase separation could apply to other proteins, or is it specific to Tg?

9. Including comparisons with other systems involving extracellular phase separation, such as insulin storage, would provide broader context and highlight the relevance of these findings.

Reviewer #2

(Remarks to the Author)

The study is extremely interesting and very well performed. The manuscript is in general well written, although I would suggest reduction of the text.

Specific comments

1) Introduction is too long. I suggest to reduce the last paragraph in which results are anticipated

2) I suggest to quote the following review when discussing about Tg endocytosis: Am J Physiol Cell Physiol. 2000 Nov;279(5):C1295-306. doi: 10.1152/ajpcell.2000.279.5.C1295

Version 1:

Reviewer comments:

Reviewer #1

(Remarks to the Author)

The authors addressed all the issues initially raised and the manuscript, in my opinion, is now ready for publication.

Extracellular phase separation mediates storage and release of thyroglobulin in the thyroid follicular lumen

Point-by-point response to the referees' comments

Reviewer 1

Thyroid hormones are produced by the thyroid gland and are essential for regulating metabolism, growth, and development. Maintaining circulating thyroid hormone levels within a healthy range is critical for overall health. The thyroid gland achieves this, in part, by utilizing an extracellular storage depot of thyroglobulin (Tg), a glycoprotein precursor for thyroid hormones. However, the molecular mechanisms underlying the assembly of soluble thyroglobulin molecules into dense storage depots remain unclear. This study provides an *in vitro* biophysical analysis of thyroglobulin phase behavior, revealing that Tg undergoes ionic strength-dependent phase separation, forming liquid-like condensates at high salt and protein concentrations. The phase separation behavior is modulated by salt concentration, with higher ionic strength reducing the protein concentration required for condensation. Notably, Tg condensates are liquid-like upon formation, as evidenced by their high degree of sphericity, fusion behavior, and fluorescence recovery after photobleaching (FRAP). However, these condensates age over time, transitioning to a gel-like state with diminished molecular mobility. Remarkably, aged Tg condensates can revert to a soluble state upon dilution. The manuscript also demonstrates the presence of protein-dense Tg globules in the follicular lumen of cultured primary mouse and human thyroid follicles, which display a high degree of sphericity. This observation supports the hypothesis that Tg undergoes phase separation *in vivo*, playing a critical role in thyroid hormone homeostasis. The authors suggest that the extracellular phase separation of Tg provides a dynamic and reversible storage mechanism, linking molecular biophysics to endocrine physiology. The findings are novel and offer insights into how Tg storage and release processes are regulated without requiring membrane-bound compartments. However, the following comments should be addressed to enhance the manuscript:

1. In the condensate dissolution experiment (Fig. 3), the dissolution starts at the center. Could the authors conduct FRAP experiments with the laser focused on different regions within the condensate (as shown in Fig. 2D) to determine molecular dynamics at the center versus near the surface? This could help explain why dissolution begins at the core (and aging starts at the periphery).”

Thank you for the suggestion. We conducted FRAP experiments on Tg condensates aged at room temperature for 20 minutes before condensates became fully aged, bleaching areas near the condensate interface and centre (Response Figure 1 or Supplementary Figure S2; also noted in the main text line 167-170). We observed higher recovery at the centre of the condensate compared to the interface at all recorded time points. This indicates faster fluorescence recovery, and thus higher molecular dynamics, at the centre of aged Tg condensates relative to the interface. This provides further evidence that aging initiates at the condensate interface.

FRAP: Tg⁴⁸⁸ Centre vs. Interface

Response Figure 1. The interface and centre of partially aged Tg condensates exhibit distinct dynamics in FRAP analysis. **Top:** FRAP analysis of areas near the interface and centre of Tg condensates aged at room temperature for approximately 20 minutes. Across all recorded time points, fluorescence recovery was consistently higher at the centre compared to the interface. The curve fitting represents the averaged FRAP data collected from five aged Tg condensates, each bleached at both the interface and centre. In all cases, fluorescence recovery at the centre exceeded that at the interface across all recorded time points. Centre: plateau = 0.29, $R^2 = 0.72$, $t_{1/2} = 25.67$ s; interface: plateau = 0.19, $R^2 = 0.72$, $t_{1/2} = 20.42$ s. Error bars represent standard deviation. **Bottom:** Confocal images of a representative condensate during FRAP. Scale bars, 10 μm . Condensates were formed at 12.5 μM Tg, 300 mM KCl and 4 w/v% PEG 20k in 50mM, pH=7.5 HEPES buffer.

2. What is the size distribution of aged condensates *in vitro*? Would it be possible to measure FRAP at different positions within larger aged condensates? The example in Fig. 2E appears too small for such analysis, but larger condensates may allow more detailed spatial FRAP studies (as pointed in 1 above).

As noted in our response to comment 1, we have compared FRAP analysis on the centre and interface of larger aged Tg condensates of around $40 \pm 5 \mu\text{m}$ in diameter as shown in Response Figure 1. The size distribution of aged condensates *in vitro* depends on the initial Tg protein concentration, with higher initial Tg concentration leading to a larger dense phase fraction as shown in Fig. 2A in the main text.

3. Is there evidence of fibril-like structures forming within aged condensates (at least *in vitro*)? The depiction in Fig. 5 suggests fibrils in the interior, but direct evidence is lacking.

Thank you for raising this point. We have not observed fibril-like structures forming within aged condensates, neither *in vitro* nor *ex vivo*. As a result, we have modified Fig. 5 in the main text. In the revised version (please see the figure attached below), instead of depicting fibril-like structures in aged Tg condensates, we show Tg molecules clustering together, leading to lower molecular dynamics. In newly-formed, liquid-like condensates and the outer layer of Tg storage depots composed of newly-synthesised Tg molecules, individual Tg molecules are more widely spaced and exhibit higher dynamics.

Response Figure 2. Summary scheme of phase separation in thyroglobulin (Tg) storage and release in the thyroid follicular lumen.

4. Fig. 4A shows a significant accumulation of Tg on the thyrocyte apical membrane. Could this represent newly synthesized Tg molecules? Would repeating this experiment over time reveal dynamic changes in condensate size or number, or are Tg concentrations expected to remain relatively stable at the membrane and within the condensates?

Yes, newly synthesised Tg molecules are located closest to the apical membrane of thyrocytes, making them highly accessible to chemical modification and subsequent retrieval by thyrocytes, as described in [1,2] (references 27 and 33 in the main text).

[1] Citterio, C. E., Targovnik, H. M. & Arvan, P. The role of thyroglobulin in thyroid hormonogenesis. *Nat Rev Endocrinol* 15, 323–338 (2019)

[2] Carvalho, D. P. & Dupuy, C. Thyroid hormone biosynthesis and release. *Molecular and Cellular Endocrinology* 458, 6–15 (2017)

Would repeating this experiment over time reveal dynamic changes in condensate size or number, or are Tg concentrations expected to remain relatively stable at the membrane and within the condensates?

Thyroid follicles adopt a 3D structure, and Tg molecules are stored in the extracellular follicular lumen, surrounded by thyrocytes. Currently, there is no established method for visualizing and tracking Tg molecules in cultured, live thyroids. In our immunofluorescence experiments, we fixed thyrocytes and permeabilized their cell membranes to allow antibodies conjugated with fluorescent dyes to access and stain Tg molecules within the lumen. This allowed us to visualize areas of TG condensates within the follicles, however, we were unable to monitor the dynamic changes of Tg condensates over time in the fixed thyroids.

Based on condensate literature, we hypothesize that Tg may move from the membrane towards condensates. Condensates in liquid state may fuse, thereby increasing in size and decreasing in number. We think this could be a great topic for future research, particularly comparing the dynamics in healthy or diseased situations.

5. Is there any other evidence of the physico-chemical properties of the condensates in Fig 4 besides roundness?

Thank you for your question. *In vitro*, we've studied the condensates by assessing roundness, observing merging, and performing FRAP. Given the fixed nature of the *ex vivo* condensates, only assessing roundness remains as an option. Notably, we've extensively studied the behaviour of Tg in conditions similar to those in the thyroid, matching pH, crowding / protein concentration and salt concentrations. While this is an unfortunate limitation, we believe the combination of *ex vivo* observations and *in vitro* screening sufficiently supports our conclusion that Tg undergoes ionic strength-dependent phase separation.

6. In Fig. 5, the depiction of a Tg outer layer around the condensate during the transition from a reversible gel to a storage depot is not directly supported by the data. While the *in vitro* experiments (Figs. 1–3) and *ex vivo* observations (Fig. 4) establish condensate formation and presence in the lumen, there is no direct evidence of a distinct Tg outer layer. Can the authors clarify or provide justification for this proposal?

Thank you for your comments. In Fig. 5, we propose a mechanism for Tg storage and retrieval based on our experimental results and the well-established ‘last-come, first-served’ model of Tg storage and release in previous studies. These studies [3-8] have provided extensive experimental evidence of multimerised Tg storage depots within the follicular lumen, as well as a distinct outer layer on these storage depots composed of newly synthesized Tg molecules. Specifically, images of multimerised Tg storage globules from phase contrast microscopy and transmission electron microscope demonstrate that these globules adopt a multi-layered onion-like structure and different layers have variable thickness and electron density, with the outer layer displaying a particularly low electron density [4].

[3] Citterio, C. E., Targovnik, H. M. & Arvan, P. The role of thyroglobulin in thyroid hormonogenesis. *Nat Rev Endocrinol* 15, 323–338 (2019).

[4] Berndorfer, U., Wilms, H. & Herzog, V. Multimerization of thyroglobulin (TG) during extracellular storage: isolation of highly cross-linked TG from human thyroids. *The Journal of Clinical Endocrinology & Metabolism* 81, 1918–1926 (1996).

[5] Brix, K., Qatato, M., Szumska, J., Venugopalan, V. & Rehders, M. “Thyroglobulin Storage, Processing and Degradation for Thyroid Hormone Liberation”. in *The Thyroid and Its Diseases: A Comprehensive Guide for the Clinician* (eds. Luster, M., Duntas, L. H. & Wartofsky, L.) 25–48 (Springer International Publishing, Cham, 2019).

[6] Carvalho, D. P. & Dupuy, C. Thyroid hormone biosynthesis and release. *Molecular and Cellular Endocrinology* 458, 6–15 (2017).

[7] Schneider, P. B. Thyroidal Iodine Heterogeneity: “Last Come, First Served” System of Iodine Turnover. *Endocrinology* 74, 973–980 (1964).

[8] Gérard, A.-C., Deneff, J.-F., Colin, I. M. & van den Hove, M.-F. Evidence for processing of compact insoluble thyroglobulin globules in relation with follicular cell functional activity in the human and the mouse thyroid. *Eur J Endocrinol* 150, 73–80 (2004).

7. Similarly, the mechanistic step in Fig. 5 depicting the relocation of newly synthesized Tg from the apical membrane to the condensate (via the "last come" red arrow) lacks experimental support. Wouldn't it make more sense for Tg to be recruited directly from the condensate itself, as demonstrated in the dissolution experiment (Fig. 3)? What is the functional role of this proposed Tg outer layer? Could the condensate alone serve as the storage depot?

Previous studies indicate that Tg storage and release follow the ‘last-come-first-served’ rule [3,7]. Multimerised Tg storage depots adopt layered structures [4], with newly synthesized Tg molecules residing in the outer layer of Tg storage depots, which are closest to the apical membrane and are the first to be retrieved into thyrocytes. The inner layers of Tg storage depots are more electron-dense and less dynamic. The mechanism illustrated in Fig. 5 is based on our findings regarding Tg phase behaviour, as well as prior experimental evidence.

We agree with the reviewer that expanding on this in the main text is helpful and have edited the discussion relating to Fig. 5 (line 232).

8. Do the authors believe this mechanism of extracellular phase separation could apply to other proteins, or is it specific to Tg?

Thank you for your question. We share the reviewer's interest in exploring other examples of extracellular phase separation. Through our literature review, we identified a study examining its role in the assembly of extracellular matrix proteins [9]. In this context, extracellular phase separation does not function as a storage mechanism. Instead, it facilitates the formation of fibrillar matrices (reference 62 in the main text).

[9] Muiznieks, L. D., Sharpe, S., Pomès, R., & Keeley, F. W., Role of Liquid-Liquid Phase Separation in Assembly of Elastin and Other Extracellular Matrix Proteins, *J Mol Biol*, 430, 4741–4753 (2018)

9. Including comparisons with other systems involving extracellular phase separation, such as insulin storage, would provide broader context and highlight the relevance of these findings.

Thank you for your suggestion. Previous studies on insulin storage demonstrate that phase separation occurs intracellularly ([10]; reference 49 in the main text). Chromogranin (CG) proteins undergo liquid-liquid phase separation in the lumen of the trans-Golgi network (TGN). Proinsulin is recruited as a client into the CG condensates inside the cell. This enables sorting and packaging of proinsulin from the Golgi apparatus to secretory storage granules before exocytosis.

In response to the reviewer's suggestion, we have addressed the role of extracellular phase separation in the assembly of extracellular matrix proteins (as mentioned in the response to comment 8) in the final paragraph of the discussion section (line 254-255; reference 62).

[10] Parchure, A. et al. Liquid-liquid phase separation facilitates the biogenesis of secretory storage granules. *Journal of Cell Biology* 221, e202206132 (2022)

Reviewer 2:

The study is extremely interesting and very well performed. The manuscript is in general well written, although I would suggest reduction of the text.

1. Introduction is too long. I suggest to reduce the last paragraph in which results are anticipated.

We sincerely thank the reviewer for taking the time to review and appreciate our work. Thank you for the positive comments. We have reduced the introduction as suggested (line 78-88).

2. I suggest to quote the following review when discussing about Tg endocytosis: Am J Physiol Cell Physiol. 2000 Nov;279(5):C1295-306. doi: 10.1152/ajpcell.2000.279.5.C1295.

We thank the reviewer for the suggestion. We have taken the suggestion and included this review as a reference when discussing about Tg endocytosis (reference 34; line 58).